# Validation of a Brief Screening Instrument for Chemical Intolerance in a Large U.S. National Sample

**DOI:** 10.3390/ijerph18168714

**Published:** 2021-08-18

**Authors:** Raymond F. Palmer, Tatjana Walker, David Kattari, Rudy Rincon, Roger B. Perales, Carlos R. Jaén, Carl Grimes, Dana R. Sundblad, Claudia S. Miller

**Affiliations:** 1Department of Family and Community Medicine, University of Texas Health Science Center San Antonio, 7703 Floyd Curl Drive, San Antonio, TX 78229, USA; walkert2@livemail.uthscsa.edu (T.W.); rinconr@uthscsa.edu (R.R.); Peralesr@uthscsa.edu (R.B.P.); Jaen@uthscsa.edu (C.R.J.); Millercs@uthscsa.edu (C.S.M.); 2Hayward Score, Carmel, CA 93921, USA; dkattari13@gmail.com (D.K.); grimes@haywardscore.com (C.G.); dana.r.sundblad@gmail.com (D.R.S.)

**Keywords:** chemical intolerance, drug intolerance, food intolerance, QEESI, BREESI, multiple chemical sensitivity, toxicant-induced loss of tolerance, prevalence

## Abstract

Background: Chemical intolerance (CI) is characterized by multisystem symptoms triggered by low levels of exposure to xenobiotics including chemicals, foods/food additives, and drugs/medications. Prior prevalence estimates vary from 8–33% worldwide. Clinicians and researchers need a brief, practical screening tool for identifying possible chemical intolerance. This large, population-based study describes the validation of a three-item screening questionnaire, the Brief Environmental Exposure and Sensitivity Inventory (BREESI), against the international reference standard used for assessing chemical intolerance, the Quick Environmental Exposure and Sensitivity Inventory (QEESI). Methods: More than 10,000 people in the U.S. responded to the BREESI and the QEESI in a population-based survey. We calculated the overall prevalence of CI in this sample, as well as by gender, age, and income. Common statistical metrics were used to evaluate the BREESI as a screener for CI against the QEESI. Results: The prevalence estimate for QEESI-defined chemical intolerance in the U.S. was 20.39% (95% CI 19.63–21.15%). The BREESI had 91.26% sensitivity (95% CI: 89.20–93.04%) and 92.89% specificity (95% CI: 91.77–93.90%). The positive likelihood ratio was 12.83 (95% CI: 11.07–14.88), and the negative likelihood ratio was 0.09 (95% CI: 0.08–0.12). Logistic regression demonstrates that the predicted probability of CI increased sharply with each increase in the number of BREESI items endorsed (Odds Ratio: 5.3, 95% CI: 4.90–5.75). Conclusions: Chemical intolerance may affect one in five people in the U.S. The BREESI is a new, practical instrument for researchers, clinicians, and epidemiologists. As a screening tool, the BREESI offers a high degree of confidence in case ascertainment. We recommend: screen with the BREESI, confirm with the QEESI.

## 1. Introduction

Chemical Intolerance: International concern over intolerances to chemicals [1,2], foods [3,4], and drugs [5] is increasing. Up to one-quarter of the U.S. population report being either “especially” or “unusually” sensitive to certain chemicals [6]. Population-based surveys in several countries estimate CI prevalence to range between 8% and 33% [2,6,7,8]. Katerndahl et al. [9] found that 20% of patients in a university family medicine clinic reported chemical intolerances. At least one in ten US adults have well-documented food allergies, and one in five report food intolerances [10,11]. A large US electronic medical records study showed that 2.1% of health plan patients reported three or more drug intolerances [12]. Similarly, a UK medical records study showed that among more than 25,000 inpatients with documented drug intolerances, 4.9% had Multiple Drug Intolerance Syndrome, defined as 3 or more adverse reactions to drugs, suggesting cross-intolerances [13].

The Quick Environmental Exposure and Sensitivity Inventory (QEESI), the most widely used clinical screening test and research tool for identifying chemical intolerance (CI), has emerged as an international reference standard (see Table 1). Table 1 shows 72 peer-reviewed journal articles using the QEESI in 16 countries with a total of over 32,000 respondents. Complete references, along with the URLs for each citation, are provided in the Appendix A file.

Although the 50-item QEESI can be completed in less than 15 min, clinicians and researchers need a more rapid way to screen for CI, in part because of clinical time constraints and respondent burden [14]. In response, we developed the “Brief Environmental Exposure and Sensitivity Inventory” (BREESI), comprised of three questions derived from the QEESI. We previously published the BREESI’s validity metrics for a sample of 293 individuals from a primary care clinic [14]. In that sample, the BREESI showed excellent positive and negative predictive values (97% and 95% respectively) and good specificity and sensitivity (90% and 87% respectively) as compared with QEESI defined results.

### The Brief Environmental Exposure and Sensitivity Inventory (BREESI)

Instructions: Please answer these three questions by checking Yes or No.

Do you feel sick when you are exposed to tobacco smoke, certain fragrances, nail polish/remover, engine exhaust, gasoline, air fresheners, pesticides, paint/thinner, fresh tar/asphalt, cleaning supplies, new carpet, or furnishings? By sick, we mean headaches, difficulty thinking, difficulty breathing, weakness, dizziness, upset stomach, etc?_Yes _NoAre you unable to tolerate or do you have adverse or allergic reactions to any drugs or medications (such as antibiotics, anesthetics, pain relievers, X-ray contrast dye, vaccines, or birth control pills) or to an implant, prosthesis, contraceptive chemical or device, or other medical/surgical/dental material or procedure?_Yes _NoAre you unable to tolerate or do you have adverse reactions to any foods such as dairy products, wheat, corn, eggs, caffeine, alcoholic beverages, or food additives (such as MSG, food dye)?_Yes _No

These three questions help gauge an individual’s tendency to react adversely to diverse substances representing three major exposure categories (chemicals, foods, and drugs) covered by the Quick Environmental Exposure and Sensitivity Inventory (QEESI).

In this manuscript, we present: (1) a new prevalence estimate for chemical intolerance in a large U.S. sample, (2) the BREESI’s screening performance in a much larger, non-clinical, U.S. population-based survey of more than 10,000 individuals, (3) an additional random cohort of 1000 Americans for the purposes of a comparative sensitivity analysis. Here, we provide a detailed evaluation of the BREESI’s sensitivity and specificity, positive and negative predictive values, and likelihood ratios.

## 2. Materials and Methods

We surveyed US adults ages 18 and older, between 1 June and 2 June 2020, using the SurveyMonkey Audience platform [15]. A description of how respondents were recruited is available at www.surveymonkey.com/mp/audience accessed on 13 August 2021. The 10,981 respondents were randomly selected from nearly 3 million online users of the SurveyMonkey platform. The survey had an abandonment rate of 10.07% and took an average of 4 min 51 s to complete. The modeled error estimate for this survey was +/− 1.37%. The data was weighted for the population sizes of the 50 states plus the District of Columbia, gender, age, race, and income within each census region to match the Census Bureau’s 2015 American Community Survey (ACS) targets.

We included a secondary cohort to further investigate the validity of the BREESI in this large sample. One thousand respondents were randomly recruited by email through the market research company Dynata, a panel (survey) company that provides recruitment services for researchers (www.dynata.com accessed on 13 August 2021). Dynata adheres to the ESOMAR market research code of conduct. Respondents were recruited from Dynata’s nationally representative research panel in each country. The sample was stratified with roughly equal numbers of participants (*n* = 1000) across seven age bands: 18–19, 20–29, 30–39, 40–49, 50–59, 60–69, and 70 and older. It was also stratified by gender for approximately equal numbers of males and females. Both cohorts responded to the exact same survey as described below, and data were analyzed in the same manner. QEESI and BREESI Scores: The QEESI has 4 scales: Chemical Exposures, Other Exposures, Symptoms, and Life Impact. There is also a 10-item Masking Index which gauges ongoing exposures, such as caffeine, alcohol, or tobacco use, that can affect individuals’ awareness of their intolerances as well as the intensity of their responses to environmental exposures [16,17]. Each scale contains 10 items which are rated from 0 to 10: 0 = “not at all a problem” to 10 = “severe/disabling symptoms.” Scale totals range from 0–100.

There are three QEESI classifications for CI, based on responses to the Chemical Exposures and Symptom Scales. Scores greater than or equal to 40 on both scales are *very suggestive* of CI. Scores from 20–39 on one or both scales are *suggestive* of CI. Scores less than 20 on both scales are *not suggestive* of CI [16,17]. We use these criteria in this study.

We derived the BREESI’s three questions from the chemical, food, and drug items on the QEESI. We compressed the ten chemical exposure items into a single “yes” or “no” question. We summarized the food and drug intolerance items on the QEESI Other Exposures Scale in two “yes” or “no” questions. Our goal was to create a brief but sensitive instrument for assessing CI in clinical settings and epidemiological/research investigations.

Statistical Analysis: We prepared CI prevalence estimates for the entire sample by gender, age, and household income. We calculated standard metrics for testing the validity of a screening instrument [18], namely: sensitivity and specificity, positive and negative predictive values, and likelihood ratios for the BREESI items against the established QEESI categories of *very suggestive* versus *not suggestive*.

Specificity, sensitivity, positive predictive values (PPV), and negative predictive values (NPV) are measures that depend upon the prevalence of the clinical event in the population under study [19]. On the other hand, positive likelihood ratios (PLR) and negative likelihood ratios (NLR) do not depend on disease prevalence and are therefore preferred and considered more accurate than NPV and PPV [20]. Therefore, we calculated the likelihood ratios values [21] as:Positive likelihood ratio (PLR): the ratio between the probability of a positive test result given the presence of the disease and the probability of a positive test result given the absence of the disease.PLR = True positive rate/False positive rate = Sensitivity/(1-Specificity)Negative likelihood ratio (NLR): the ratio between the probability of a negative test result given the presence of the disease and the probability of a negative test result given the absence of the disease.NLR = False negative rate/True negative rate = (1-Sensitivity)/Specificity

A PLR greater than 10 is strong evidence for determining a disease condition is present. Conversely, a NLR less than 0.10 is strong evidence for ruling out a disease condition [22,23]. The accuracy statistic (e.g., the receiver operator curve) indicates an overall performance of the test.

Using logistic regression, we also determined Odds Ratios (OR) with 95% confidence intervals and the c-statistic for the BREESI as a predictor of CI (*very suggestive* vs *not suggestive)*. Potential confounding variables were included in a multivariate model. All analyses were conducted using SAS statistical software [24].

## 3. Results

Sample demographics: Among the 10,981 survey respondents, 257 did not complete the QEESI Chemical Exposures or Symptom Scales and were removed from the analysis. Demographics for the 10,724 respondents with complete QEESI data appear in Table 2. To evaluate the sample coverage, we compare our demographics with the American Community Survey (ACS) [25,26]. The final demographic sample we derived was very close (within approximately 10%) to the estimates obtained by the ACS. Our sample has a slight gender and younger participant bias as well as a slightly lower percentage of household incomes over $100,000.

As shown in Table 2, our CI prevalence estimate for this sample, based on the QEESI *very suggestive* category, is 20.39%. The secondary validity sample yielded a slightly higher prevalence rate of 25.00% (not shown in the table).

Table 3 shows CI prevalence rates by gender, age, and household income. Females were significantly more likely than males to report CI (22.6% vs. 18.21%, *p* < 0.001). Respondents older than 60 years of age were significantly less likely to have scores *very suggestive* of CI compared to those 60 years of age or younger (approximately 11 % vs. 23%, *p* < 0.001). Those in the lower-income categories were more likely to have scores *very suggestive* of CI than respondents reporting $100,000 or more a year, or those who preferred not to report their household income. Average QEESI scale scores for each QEESI group (*not suggestive* of CI, *suggestive* of CI, and *very suggestive* of CI) are also shown in Table 3. As would be expected, scores of the *not suggestive* group are very much lower than the other groups, with the *very suggestive* group having the highest scores on the Chemical Exposures and Symptom Scales (*p* < 0.001).

BREESI responses: One-third of participants responded “No” to all three BREESI items (*n* = 3614, 33.7%). Approximately one-third (*n* = 3320, 30.96%) chose one item only; 21.55% (*n* = 2311) chose two items; and 13.79% (N = 1479) chose all three. The Venn diagram in Figure 1 shows the overlap and percentage of those who chose at least one BREESI item (*n* = 7110 or 66% of the total sample). The chemical item accounts for most of the variability. Twenty-one percent of those choosing any item chose all three items.

The logistic regression probability graph in Figure 2 is consistent with the bar graph trend in Figure 3. The predicted probability of CI increases sharply as more BREESI items are endorsed. With no BREESI items selected, the probability of CI is about 5%; with one BREESI item, 32%; two items, 75%; and with all three items, 90%. Thus, the odds of CI increase with each additional BREESI item chosen (OR = 5.3, 95% confidence interval = 4.90–5.75, ROC = 0.87). Adjusting for age, gender, and income in the logistic model did not alter the outcome (OR = 5.4, 95% confidence interval = 5.00–5.90, ROC = 0.89). Similarly, the secondary validity sample demonstrated the same Odds Ratio as the primary sample (OR = 5.22, 95% CI = 3.93–6.92, ROC = 0.87).

The metrics in Table 4 indicate how well the BREESI correctly categorizes those with, and without QEESI identified chemical intolerance. The performance metrics are given for the primary sample and for the secondary validity sample. The sensitivity indicates how well a test predicts true positive cases. Specificity indicates how well a test predicts true negative cases. In the primary sample of 10,000, the BREESI showed a sensitivity of 91% and specificity of 93%. Positive predictive value (PPV) is the probability that subjects with a positive screening test truly have the condition. A negative predictive value (NPV) is the probability that subjects with a negative screening test do not have the condition. The PPV for the BREESI was 83% accurate in predicting CI. The NPV indicates that the BREESI was 97% accurate in classifying those without CI. The accuracy statistic was 92% accurate for the BREESI as a predictor of CI.

In the secondary random sample of 1000 Americans, shown at the bottom of Table 4, the BREESI’s overall performance metrics were comparable to that of the larger sample. By these statistical metrics, both samples provide evidence that the BREESI performs well as a screening tool for CI.

## 4. Discussion

Our earlier study of 293 primary care patients [14] showed that the BREESI exhibited good positive and negative predictive values, as well as sensitivity and specificity, when evaluated against the QEESI reference standard [14]. This suggested that the BREESI might be an efficient tool for determining potential high likelihood CI, but one requiring evaluation in other larger population samples. The results of this U.S. population-based study in more than 10,000 individuals confirm the BREESI’s performance, based upon the same predictive performance metrics as in the previous study. To address concerns about prevalence estimates affecting the Sensitivity, Specificity, Positive and Negative Predictive Values, we included the Positive and Negative Likelihood Ratios, which are not influenced by disease prevalence. Table 4 shows that all performance metrics were excellent, confirming the BREESI as an efficient and reliable chemical intolerance screening tool in the U.S. population. We recommend using a BREESI cutoff score of “3” for epidemiological studies due to its high congruence with CI. As shown in Figure 2, if all three items are chosen, there is a 91% likelihood that QEESI classification would be *very suggestive* of CI. On the other hand, in the clinical setting, even a score of “1” on the BREESI yields a 36% change of *very suggestive* of CI. To avoid the possibility of missing a case, a BREESI score of 1 should prompt clinicians to administer the QEESI.

Potential uses of the BREESI: The BREESI is not a substitute for the QEESI, but rather a time-saving tool to identify individuals with potential CI in medical clinics or epidemiological studies. Identifying those who are, or are not, likely to have CI simply by asking the three BREESI questions can reduce clinical assessment time. Researchers, clinicians, health plans, epidemiologists, and others can use the BREESI to screen for chemical intolerance. Individuals who endorse any of the BREESI items should take the full QEESI to help identify specific chemical, food, and drug triggers. Including everyone who answers yes to any one of the three BREESI items makes it unlikely that an individual with CI will be overlooked. Individuals who endorse two or three BREESI items have an even greater likelihood of meeting CI criteria. Using the BREESI, followed by the QEESI, enables practitioners to identify patients who are more chemically intolerant so they may be counseled to avoid or minimize their exposures. New-onset (or marked worsening) of chemical, food, and/or drug intolerances is a hallmark of chemical intolerance, much as fever signifies possible infection. For clinical studies, we would suggest administering the BREESI to patients at clinical visits, just as medications and allergies are assessed and updated. The finding of *very suggestive* of CI is an important tool in the diagnostic process but does not by itself establish the diagnosis of CI and should not be an end, but an important part of the process of diagnosis and potential intervention, particularly given the known cluster of other treatable conditions that often accompany CI.

For large-scale epidemiological studies, the BREESI score of three would capture CI with a relatively high degree of confidence. Depending on the size and scope of larger studies, researchers could consider a BREESI score of two to capture more cases of CI.

CI prevalence: The prevalence results of our primary and secondary samples reported in this study support those of previous studies showing a high prevalence of self-reported intolerances to chemicals, foods, and drugs in the U.S. population—although the range has been estimates between 8 and 33%. The prevalence of *very suggestive* CI based on QEESI responses in this population-based sample of more than 10,000 people in the U.S. was 20.39% (95% CI 19.63–21.15%) and 25% (95% CI 22.1–28.0) in the secondary sample. We believe this is the largest study to date to estimate CI prevalence in the U.S. Based on the 2019 U.S. Census Bureau population estimate [26], our study suggests that more than 50 million U.S. adults have chemical intolerance.

Limitations: Our data come from a national survey platform limited to those with access to computers. The digital divide (e.g., opportunities to access information and communication technologies through the Internet) has continued to narrow over the last 20 years with recent studies indicating that age and gender are less affected, but low income remains associated with lower access [27]. As indicated in Table 2, our sample was approximately 10% lower in the income bracket of over $100,000 compared to the ACS. Despite these potential access biases, the primary justification for using online platforms is that they are quick, affordable, and yield a sufficiently large sample to be representative of a reasonable population estimate. Clinical samples would only be representative of the region they serve, and even then, only of the sick population. Phone surveys are significantly more expensive and time intensive.

## 5. Conclusions

CI is an underappreciated driver of morbidity among patients who may not know how to report their symptoms to their doctors or on health questionnaires. Despite the high prevalence of CI, clinicians and/or other researchers may fail to diagnose or identify CI because they do not know the pertinent questions to ask. Researchers and epidemiologists may be missing important opportunities to understand CI as it relates to other conditions or research contexts. The three-item BREESI makes it possible to assess CI rapidly and with a fairly high degree of confidence and should, when warranted, prompt a more comprehensive assessment using the QEESI. We recommend: screen with the BREESI, confirm with the QEESI.

## Figures and Tables

**Figure 1 ijerph-18-08714-f001:**
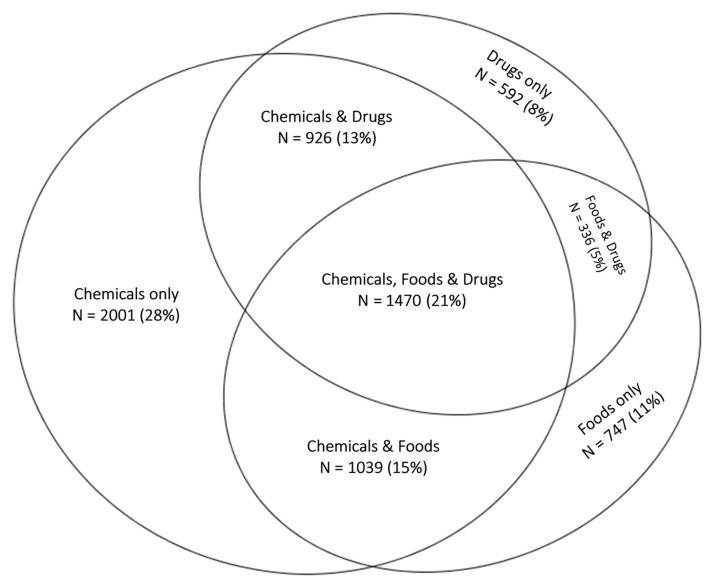
Venn diagram depicting overlap between BREESI responses of one or more items (*n* = 7110).

**Figure 2 ijerph-18-08714-f002:**
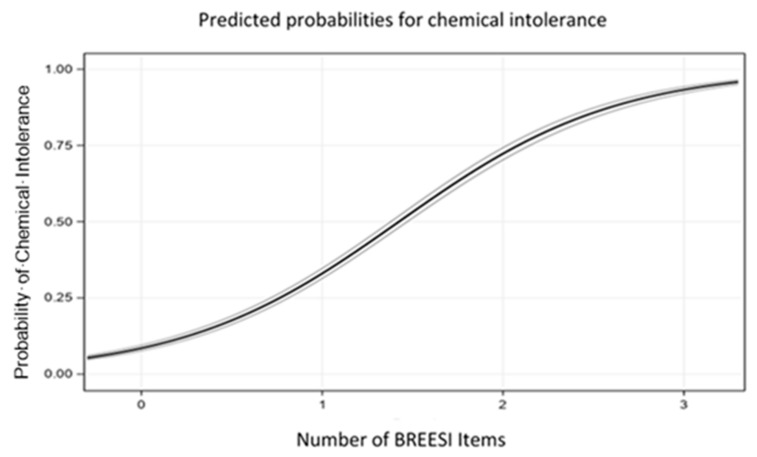
Predicted probability of chemical intolerance versus number of BREESI items endorsed. BREESI statistical validity performance.

**Figure 3 ijerph-18-08714-f003:**
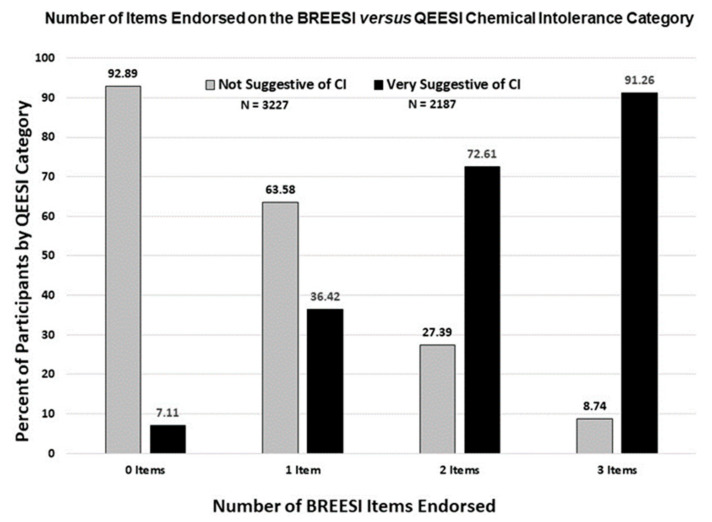
Number of BREESI items endorsed for low (*not suggestive*) versus high (*very suggestive*) QEESI chemical intolerance categories. Figure 2 shows that as more BREESI items are chosen, the percentage having QEESI-defined CI (solid black bars) increases. As fewer BREESI items are chosen, a much lower percentage of *very suggestive* of CI is observed.

**Table 1 ijerph-18-08714-t001:** Peer-reviewed Journal Articles Using the QEESI by Country.

Country	Author & Date	N	Country	Author & Date	N
Austria	Weiss, 2017	72	Japan	Hojo, 2009	412
China	Huang, 2011	658	Japan	Hojo, 2018	214
China	Huang, 2014	658	Japan	Hojo, 2019	555
Columbia	Storino, 2021	1	Japan	Ishibashi, 2007	214
Denmark	Dantoft, 2014	298	Japan	Lu, 2020	667
Denmark	Hauge, 2015	69	Japan	Manabe, 2008	368
Denmark	Skovbjerg, 2012	1493	Japan	Mizuki, 2004	32
Denmark	Tran, 2014	3	Japan	Mizuki, 2015	40
Denmark	Tran, 2017	39	Japan	Mizukoshi, 2015	8
Finland	Heinonen-Guzejev, 2012	327	Japan	Nakaoka, 2018	43
Finland	Selinheimo, 2019	52	Japan	Ohsawa, 2020	2
Finland	Vuokko, 2019	12	Japan	Suzuki, 2020	141
France	Kamoun, 2011	20	Japan	Watai, 2018	528
Germany	Bauer, 2007	202	Japan	Yoshino, 2004	69
Germany	Schnakenberg, 2007	521	Saudi Arabia	Khalil, 2020	134
Indonesia	Hildebrandt, 2019	471	South Korea	Heo, 2017	1030
Indonesia	Kubota,2020	707	South Korea	Jeon, 2012	300
Italy	Caccamo, 2013	443	South Korea	Jeong, 2014	379
Italy	De Luca, 2010	444	South Korea	Yun, 2013	1
Italy	De Luca, 2014	300	Spain	Aguilar-Aguilar, 2018	52
Italy	De Luca, 2015	563	Spain	Alobid, 2014	118
Italy	Gugliandolo, 2016	34	Spain	Fernandez-Solà, 2005	75
Italy	Micarelli, 2016a	38	Spain	García-Sierra, 2014	125
Italy	Micarelli, 2016b	38	Spain	Lago Blanco, 2016	73
Italy	Viziano, 2017	38	Spain	Mena, 2013	231
Japan	Azuma, 2013	23	Spain	Nogué, 2007	52
Japan	Azuma, 2015a	7245	Spain	Paredes-Rizo, 2018	1
Japan	Azuma, 2015b	12	Spain	Pérez-Crespo, 2018	514
Japan	Azuma, 2016	16	Spain	Aguilar-Aguilar, 2018	52
Japan	Azuma, 2019	909	Sweden	Andersson, 2009	207
Japan	Cui, 2013	324	Sweden	Nordin, 2010	283
Japan	Cui, 2014	2464	United States	Gould Peek, 2015	563
Japan	Cui, 2015	565	United States	Heilbrun, 2015	694
Japan	Fujimori, 2012	1084	United States	Katerndahl, 2012	400
Japan	Hasegawa, 2009	51	United States	Miller, 1999a	421
Japan	Hojo, 2002	1260	United States	Miller, 1999b	421
Japan	Hojo, 2003	760	United States	Palmer, 2020	293
Japan	Hojo, 2005	440	Uruguay	De Ben, 2014	2
Japan	Hojo, 2008	106	16 Countries Total N > 32,000 subjects		

**Table 2 ijerph-18-08714-t002:** Sample demographics (N = 10,724) Compared to the American Community Survey.

		N	Percentage of Sample	Percentage in ACS
Gender	Male	4927	53.36%	48.68%
	Female	5636	46.64%	51.32%
	Missing	161	1.50%	
Age				
	18–29	2788	26.39%	13.10%
	30–44	2320	21.96%	27.69%
	45–60	3143	29.75%	26.87%
	>60	2312	21.89%	32.34%
	Missing	161	1.50%	
Household Income/Year				
	<$25,000	1839	17.45%	18.1%
	$25,000–$49,999	2018	19.15%	20.3%
	$50,000–$74,999	1965	18.65%	17.4%
	$75,000–$99,999	1358	12.89%	12.8%
	≥$100,000	2341	22.21%	31.4%
	Prefer not to answer	1018	9.66%	
	Missing	185	1.7%	
QEESI Score				
	Not Suggestive of CI	3227	30.09%	NA
	Suggestive of CI	5310	49.52%	NA
	Very Suggestive of CI	2187	20.39%	NA
Number of BREESI Items Endorsed				
	0	3614	33.7%	NA
	1	3320	30.96%	NA
	2	2311	21.55%	NA
	3	1479	13.79%	NA

**Table 3 ijerph-18-08714-t003:** Chemical intolerance prevalence by demographic group and mean scale scores (N = 10,724).

		Not Suggestive of CI	Suggestive of CI	Very Suggestive of CI
Total Sample		30.09%	49.52%	20.39%
Gender	Male	36.76%	45.04%	18.21%
	Female	23.94%	53.46%	22.6% ***
Age	18–29	26.72%	49.28%	24.00%
	30–44	26.9%	49.83%	23.28%
	45–60	27.39%	50.30%	22.3%
	>60	40.22%	48.49%	11.29% ***
Household Income/Year				
	<25,000	27.35%	47.85%	24.8%
	$25,000–<$50,000	25.82%	51.34%	22.84%
	$50,000–<$75,000	27.94%	50.08%	21.98%
	$75,000–<$100,000	29.68%	49.63%	20.69%
	>$100,000	34.39%	48.83%	16.79% ***
	Prefer not to answer	36.25%	49.41%	14.34%
QEESI Scale Scores	Total	Not Suggestive of CI	Suggestive of CI	Very Suggestive of CI
	Mean (SD)	Mean (SD)	Mean (SD)	Mean (SD)
Chemical Exposures Score	28.50 (22.49)	6.10 (5.90)	29.91 (16.12)	58.14 (13.17) ***
Symptom Score	28.58 (22.10)	6.97 (5.94)	29.34 (15.13)	58.64 (13.56) ***

Mantel-Hansel Chi-Square *** *p* < 0.001, comparing *Very Suggestive* within demographic categories.

**Table 4 ijerph-18-08714-t004:** Statistical performance metrics of the BREESI.

	Value	95% Confidence Interval
United States sample (*n* = 10,724)		
Sensitivity	91.26%	89.20% to 93.04%
Specificity	92.89%	91.77% to 93.90%
Positive Predictive Value	82.89%	80.68% to 84.89%
Negative Predictive Value	96.57%	95.79% to 97.21%
Positive Likelihood Ratio	12.83	11.07 to 14.88
Negative Likelihood Ratio	0.09	0.08 to 0.12
Accuracy	92.44%	91.47% to 93.33%
Additional United States Sensitivity sample (N = 1000)		
Sensitivity	78.71%	71.42% to 84.87%
Specificity	97.28%	93.18% to 99.25%
Positive Predictive Value	92.85%	83.13% to 97.17%
Negative Predictive Value	91.05%	88.24% to 93.23%
Positive Likelihood Ratio	28.93	10.96 to 76.31
Negative Likelihood Ratio	0.22	0.16 to 0.30
Accuracy	91.52%	87.79% to 94.41%

## Data Availability

The dataset analyzed during the current study is available from the corresponding author on reasonable request.

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
