# Peer review of "Validation of a Brief Screening Instrument for Chemical Intolerance in a Large U.S. National Sample"

_ijerph, 2021, doi:10.3390/ijerph18168714_

Round 1
Reviewer 1 Report
Summary
This manuscript validated a brief and practical screening tool, the Brief Environmental Exposure and Sensitivity Inventory (BREESI), to help detect chemical intolerance (CI) in a US population-based survey. The validation was against the international reference standard used for assessing CI, the Quick Environmental Exposure and Sensitivity Inventory (QEESI). Descriptive analysis of overall prevalence by demographics showed the BREESI had 91.26% sensitivity and 92.89% specificity, and logistic regression demonstrated high odds ratio (5.3) for BREESI items numbers. It concluded CI might affect one in five US people; the BREESI offers a high degree of confidence in CI detection.
Broad comments
Strength: The study has a novelty in topic. BREESI is a novel tool in CI detection, and few studies have validated it in larger/other populations. This study validated the BREESI’s performance in a larger US population.
Weakness: The study lacked some sensitivity analyses. Compared with QEESI, which has widespread use in many countries, BREESI is novel and was developed on a 293 sample size; it still needs validation in larger/other populations (Palmer, Raymond F., et al., 2020). Since there were nearly three million online users, some sensitivity analyses can be considered to enhance the validity of BREESI. More validations such as using different sample sizes randomly sampled from the online users or using random sampling stratified by states could further help justify the BREESI in the US population.
Specific comments
In lines 211-213, an explanation on what is “cutoff score of ‘3’” or “a score of ‘1’” for BREESI is needed. It might be cases when we have “Yes” to three or one BREESI question. But it could be clearer to readers when there is an explanation.
Some typos should be handled, such as in line 197 ‘good screening qualities.3.2’.
Author Response
Summary
This manuscript validated a brief and practical screening tool, the Brief Environmental Exposure and Sensitivity Inventory (BREESI), to help detect chemical intolerance (CI) in a US population-based survey. The validation was against the international reference standard used for assessing CI, the Quick Environmental Exposure and Sensitivity Inventory (QEESI). Descriptive analysis of overall prevalence by demographics showed the BREESI had 91.26% sensitivity and 92.89% specificity, and logistic regression demonstrated high odds ratio (5.3) for BREESI items numbers. It concluded CI might affect one in five US people; the BREESI offers a high degree of confidence in CI detection.
Broad comments
Strength: The study has a novelty in topic. BREESI is a novel tool in CI detection, and few studies have validated it in larger/other populations. This study validated the BREESI’s performance in a larger US population.
Weakness: The study lacked some sensitivity analyses. Compared with QEESI, which has widespread use in many countries, BREESI is novel and was developed on a 293 sample size; it still needs validation in larger/other populations (Palmer, Raymond F., et al., 2020). Since there were nearly three million online users, some sensitivity analyses can be considered to enhance the validity of BREESI. More validations such as using different sample sizes randomly sampled from the online users or using random sampling stratified by states could further help justify the BREESI in the US population.
Thank you for this astute comment. We agree that a sensitivity or secondary validation sample would be good to include. As such, we’ve included data from a second random US population from a recent unpublished study we’ve conducted. This is stated in line 74. A description of the new random sample is included the Materials and Methods section beginning on line 85. The performance metrics of this sample have been included in Table 4. Statistics of that sample are provided on line 135,199 and 211-225.
Specific comments
In lines 211-213, an explanation on what is “cutoff score of ‘3’” or “a score of ‘1’” for BREESI is needed. It might be cases when we have “Yes” to three or one BREESI question. But it could be clearer to readers when there is an explanation.
Thank you, this has been clarified on lines 257-260.
Some typos should be handled, such as in line 197 ‘good screening qualities.3.2’.
This, and other typos as well as grammar and clarifications have been corrected throughout the manuscript.
Reviewer 2 Report
Validation of a Brief Screening Instrument for Chemical Intollerance in a large U.S. National Sample.
By Palmer et al.
The manuscript reports the population-based study validation of the three-item screening questionnaire (Brief Screening Instrument for Chemical Intollerance -BREESI) test against the international reference standard used for assessing chemical intolerance, the Quick Environmental Exposure and Sensitivity Inventory (QEESI).
The study is relevant as in recent time the number of people affected by chemical intolerance is increasing and the application of this brief test allow to quickly recognize people affected by this disease.
The research design is fully appropriate, the methodology is correct, results are reported in clear manner, and discussion and conclusion are supported by the data not hiding the critical points of the test. The novelty respect to the previous paper (where the comparison was done with about two orders of magnitude lower number of people) is evident and clearly reported.
I strongly agree with the publication of the paper in IJERPH journal in this form.
Author Response
Reviewer 1:
Validation of a Brief Screening Instrument for Chemical Intolerance in a large U.S. National Sample. By Palmer et al.
The manuscript reports the population-based study validation of the three-item screening questionnaire (Brief Screening Instrument for Chemical Intolerance -BREESI) test against the international reference standard used for assessing chemical intolerance, the Quick Environmental Exposure and Sensitivity Inventory (QEESI).
The study is relevant as in recent time the number of people affected by chemical intolerance is increasing and the application of this brief test allow to quickly recognize people affected by this disease.
The research design is fully appropriate, the methodology is correct, results are reported in clear manner, and discussion and conclusion are supported by the data not hiding the critical points of the test. The novelty respect to the previous paper (where the comparison was done with about two orders of magnitude lower number of people) is evident and clearly reported.
I strongly agree with the publication of the paper in IJERPH journal in this form.
We thank the reviewer for their time and expertise.